

# No conclusive evidence that difficult general knowledge questions cause a "Google Stroop effect". A replication study

Guido Hesselmann

Department of General and Biological Psychology, Psychologische Hochschule Berlin (PHB), Berlin, Germany

## ABSTRACT

Access to the digital "all-knowing cloud" has become an integral part of our daily lives. It has been suggested that the increasing offloading of information and information processing services to the cloud will alter human cognition and metacognition in the short and long term. A much-cited study published in *Science* in 2011 provided first behavioral evidence for such changes in human cognition. Participants had to answer difficult trivia questions, and subsequently showed longer response times in a variant of the Stroop task with internet-related words ("Google Stroop effect"). The authors of this study concluded that the concept of the Internet is automatically activated in situations where information is missing (e.g., because we might feel the urge to "google" the information). However, the "Google Stroop effect" could not be replicated in two recent replication attempts as part of a large replicability project. After the failed replication was published in 2018, the first author of the original study pointed out some problems with the design of the failed replication. In our study, we therefore aimed to replicate the "Google Stroop effect" with a research design closer to the original experiment. Our results revealed no conclusive evidence in favor of the notion that the concept of the Internet or internet access (via computers or smartphones) is automatically activated when participants are faced with hard trivia questions. We provide recommendations for follow-up research.

## INTRODUCTION

It seems intuitively plausible that today's ubiquitous 24/7 access to the Internet via smartphones and computers will affect our cognitive functioning and strategies. Specifically, it has been suggested that different types of "cognitive offloading" (i.e., the use of our bodies, objects, and technology to alter the processing requirements of a task to reduce cognitive demand) may alter human cognition and metacognition in the short and long term (*Risko & Gilbert, 2016*). In the memory domain, one idea is that the Internet is taking the place not just of other humans as external sources of memory ("transactive memory"), but also of our own cognitive faculties (*Ward, 2013*; *Wegner & Ward, 2013*). We increasingly offload

Corresponding author
Guido Hesselmann,
g.hesselmann@phb.de,
g.hesselmann@gmail.com

[1] A total of 1,296 citations on Google Scholar, as of October 7 2020.

[2] The study is mentioned in a 2019 publication by the EU on harmful internet use: https://op.europa.eu/en/publication-detail/-/publication/fb2d58ea-8e58-11e9-9369-01aa75ed71a1/language-en/format-PDF/source-127484707.

information to "the cloud", as almost all information today is readily available through a quick Internet search.

Evidence for such "Google effects on memory" has been presented in a much-cited[1] and influential[2] paper published in *Science* (*Sparrow, Liu & Wegner, 2011*). Across four experiments, the authors showed that (a) when people expect to have future access to information, they have lower rates of recall of the information itself and enhanced recall instead for where to access it, and (b) when faced with difficult general knowledge (or, trivia) questions, people are primed to think about the Internet and computers. The latter effect was demonstrated in one experiment (Exp.1) where participants answered easy or difficult trivia questions, and then completed a variant of the Stroop task (*MacLeod, 1991*). In a paradigm conceptually similar to the emotional Stroop paradigm (*Algom, Chajut & Lev, 2004*), participants responded to the ink color of written words, which were either related or unrelated to the Internet. Stroop-like interference from words relating to computers and Internet search engines was increased after participants answered difficult compared with easy questions, consistent with those terms being "primed" in participants' minds (*Doyen et al., 2014*). Briefly, the results seem to suggest that whenever information is needed and lacking, the concept of the Internet (including computer-related terms) is activated and can interfere with our behavior in subsequent tasks (e.g., because we might feel the urge to "google" the information).

In 2018, Camerer and colleagues published a meta-analysis of 21 replications of social science and psychology experiments published in *Science* or *Nature* between 2010 and 2015 (*Camerer et al., 2018*). This large replicability project included a replication of the "Google Stroop effect" by Sparrow and colleagues (*2011*). All materials and data from this replication –led by Holzmeister & Camerer—are freely available on OSF (https://osf.io/wmgj9/). This replication tested the hypothesis that, after answering a block of hard trivia questions, color-naming reaction times (RTs) are longer for computer-related terms than for general words but did not show a significant effect despite adequate statistical power (see https://osf.io/4rfme/ for a short summary of this replication). However, as the original authors of *Sparrow, Liu & Wegner (2011)* did not provide *Camerer et al. (2018)* with any materials or feedback on their inquiries, it was difficult to replicate the experimental design of the original study. After the replication had been completed and published, Sparrow noted some design differences compared to the original study (*Sparrow, 2018*). As the authors point out (*Camerer et al., 2018*), it cannot be ruled out that these design differences, including the manipulation of cognitive load (see below), influenced the replication result. Therefore, it was our aim to investigate the "Google Stroop effect" in a further study, based on the original experiment, the materials provided by Holzmeister & Camerer, as well as the critical comments by the original authors.

## MATERIALS & METHODS

### Sample

This work is based on an undergraduate student research project ("Experimentell-Empirisches Praktikum—ExPra") at the Psychologische Hochschule Berlin (PHB). A total

[3]Our own request for further details about the experimental design remained unanswered (email from March 2019).

of 117 participants were tested. The sample consisted of students at the Psychologische Hochschule Berlin (PHB), as well as friends and families of the student experimenters (see acknowledgment). All participants provided written informed consent. The experiment was approved by the ethics committee of the PHB (approval number PHB10032019).

## Paradigm

Our version of the experiment was based on two previous studies: Sparrow et al.'s original Exp.1 (*Sparrow, Liu & Wegner, 2011*), and the replication study (*Camerer et al., 2018*). We incorporated comments provided by the first author of the original study (*Sparrow, 2018*), published in response to the failed replication[3].

The original experiment tested the hypothesis that participants are primed to think about the Internet when faced with difficult trivia questions (e.g., "Did Benjamin Franklin give piano lessons?"). Participants first had to answer a block of either hard or simple question followed by a modified Stroop Task. In this task, Internet-related and neutral words were presented in random order, and participants were instructed to indicate the word's color (blue or red) via button press. RTs to the words were measured as the dependent variable. In order to manipulate cognitive load, a random six-digit number was presented, and participants were instructed to memorize it for delayed retrieval (Fig. 1). After their response in the modified Stroop Task, participants were asked to enter the six-digit number. The results from the modified Stroop task showed the predicted pattern: RTs to computer terms (e.g., Google) were longer than RTs to neutral terms (e.g., Target), especially after participants were faced with difficult trivia questions ("question type $\times$ word type" interaction; $F(1,66) = 5.02$, $p < .03$).

Although *Sparrow, Liu & Wegner (2011)* reported that they used eight target words related to computers and search engines, and 16 unrelated words, the total number of trials presented in each "hard question" and "easy question" block remained unclear, and also whether words were repeated. In their replication, *Camerer et al. (2018)* used the 24 words originally reported by *Sparrow, Liu & Wegner (2011)*, and decided to run 48 trials per block, resulting in the repetition of words (i.e., each participant saw each word four times, twice in the "hard question" block, and twice in the "easy question" block.) In her response to the failed replication (*Sparrow, 2018*), Sparrow then strongly argues against the repetition of words, and reported the full set of 16 Internet-related words used in the original study.

The cognitive load manipulation is described as follows in the original study: "Participants are presented with words in either blue or red, and were asked to press a key corresponding with the correct color. At the same time, they were to hold a 6 digit number in memory, creating cognitive load" (*Sparrow, Liu & Wegner, 2011*, supplement). In their replication, *Camerer et al. (2018)* decided to manipulate cognitive load by presenting a six-digit number before each Stroop task block. After the block (involving several trials), participants were asked to enter the memorized number. In her response, Sparrow strongly argues against this block-wise procedure and provided more detail about the original study: "[…] before each word, they were shown a different six digit number, which was reported back by them after each Stroop word".
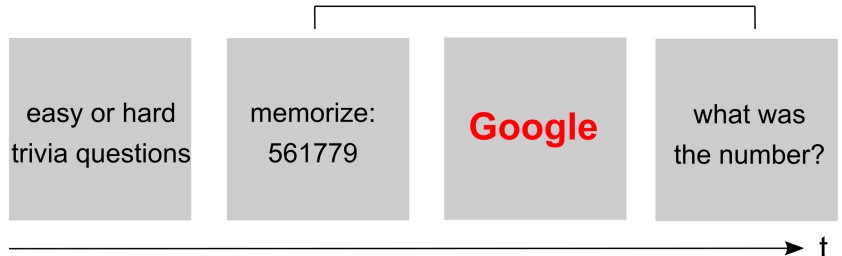

**Figure 1  Schematic illustration of the modified Stroop paradigm.** Participants first answer a list of 16 easy or hard trivia questions, followed by a block of color-naming trials (modified Stroop task). Before each Stroop block, or before each Stroop trial, a random six-digit number is presented for later retrieval (i.e., delayed retrieval at the end of a block of several Stroop trials, or at the end of each Stroop trial).

Finally, Sparrow mentions that the original experiment was run in 2006, which is why she believes that the computer terms used originally are obsolete and should not have been used in a replication (*Sparrow, 2018*). In her response, she writes that she "would focus primarily on target words about phones (as they seem most ubiquitous), but would also be sure to pre-test many possible words to ensure their contextual relevance before putting them into the modified Stroop".

For our replication study, we used the material provided by Holzmeister & Camerer on the open science platform OSF (https://osf.io/wmgj9/), and revised it according to Sparrow's comments. All materials and data are available at OSF (https://osf.io/cjgea/). The experiment was implemented using oTree (*Chen, Schonger & Wickens, 2016*). We translated the trivia questions and Stroop words into the German language, and adjusted some questions to the German context (e.g., "Was Cat in the Hat written by JD Salinger?"). The experiment consisted of two blocks of 16 either hard or easy questions, followed by 24 Stroop words of which eight were Internet-related (e.g., WLAN, Google, Website) and 16 were unrelated (e.g., frame, bottle, bamboo). The words were pre-tested for contextual relevance (see below), and randomly assigned to the easy and hard question condition. Words were presented in random order, and word color (blue or red) was randomly chosen on each trial. Participants were instructed to indicate the word color as quickly and as accurately as possible via button press ("e" for blue and "i" for red, using the index fingers of both hands). Participants were asked to place their fingers on the keys of the computer keyboard before the start of the Stroop task. Before the presentation of each Stroop word, participants had to memorize a random six-digit number for delayed retrieval after their response in the color-naming task.

After the main experiment, participants provided information about their age, level of education, color blindness, and what they thought the purpose of the experiment was. Finally, participants were asked what they would normally do when faced with a hard general knowledge question in their daily life: (a) look up the answer on the Internet, (b) ask someone who might know the answer, (c) leave it at that. Including instructions, debriefing and signing the informed consent forms, the experiment lasted approximately 45 min in total.

## Word stimuli

In the original study, the color naming task contained "8 target words related to computers and search engines (e.g., Google, Yahoo, screen, browser, modem, keys, Internet, computer), and 16 unrelated words (e.g., Target, Nike, Coca Cola, Yoplait, table, telephone, book, hammer, nails, chair, piano, pencil, paper, eraser, laser, television)" (supplement, p.2). According to *Sparrow (2018)*, the computer-related target words were selected from a larger set of the following 16 words: "Google, Yahoo, mouse, keys, Internet, browser, computer, screen, Altavista, Wikipedia, disk, Lycos, Netscape, modem, router, online" (p.1). The full list of words included in the set of general terms is not provided. Thus, at least in the case of computer terms, the number of target words used in the experiment was smaller than the number of available target words (eight and 16, respectively). It remains unclear how the target words were chosen for each participant. What seems to be clear is that the words Target, Nike, Google, and Yahoo were chosen for each participant, because a within-subject comparison was calculated using this subset of words (see below). Furthermore, the authors of the original study write that the sets "were matched for frequency to the target words (11)" (supplement, p.2). Without the full word sets, this claim is hard to evaluate. A brief search in the referenced word corpus (reference 11) revealed no hit for the computer terms "Altavista" and "Google" (*Nelson, McEvoy & Schreiber, 2004*).

Based on Sparrow's reply to the first replication (*Sparrow, 2018*), we prepared two new sets of German words, one set with general terms (32 words), and one set with computer terms (16 words). As suggested by *Sparrow (2018)*, we validated the contextual relevance of these words. 31 naïve participants (mostly undergraduate students who did not participate in the main experiment) took part in an online survey (https://www.surveymonkey.de/). For each of the 48 words, participants rated the contextual relevance (i.e., "Internet-relatedness") on a 5-point scale (statement: "I think of the Internet when reading this word"; Rating: 1 = Strongly agree; 2 = Agree; 3 = Neutral; 4 = Disagree; 5 = Strongly disagree). For computer words, the mean rating (i.e., the mean rating across the median ratings per participant) turned out to be 1.06, while it was 4.94 for the neutral terms.

Our list of computer (or, Internet-related) terms contained: website, data volume, email, search engine, Wikipedia, WLAN, app, Google, smartphone, hotspot, online, blog, Spotify, Firefox, Whatsapp, Chrome. The list of general (or, unrelated) terms contained words like nail or car, but also names of grocery stores (all words in German; full list on OSF). After completion of the study, we used the online DlexDB database to estimate word frequency in the two sets (http://www.dlexdb.de/query/kern/typposlem/). The median absolute type frequency (corpus frequency) for the general terms was 374, while it was 23.5 for the computer terms. However, half of the words in the list of computer terms were not part of the data base (Wikipedia, WLAN, Google, smartphone, blog, Spotify, Firefox, Whatsapp). To what degree this difference in word frequency between the two sets could be problematic for the current experiment, is hard to say. Under the assumption that participants read the words when responding to their color, one would expect longer RTs for words in the set of computer terms due to the lower word frequency (*Larsen, Mercer & Balota, 2006*). The main effect of "word type" reported by *Sparrow, Liu & Wegner (2011)*

[4]Using a Bayes Factor approach (https://richarddmorey.github.io/BayesFactor/), we found that a model not containing word frequency as predictor was preferred to a model containing word frequency by a factor of 16. Details can be found in the R analysis script on OSF

could parsimoniously be explained by differences in word frequency, if word sets were not matched. However, differences in word frequency between the two sets do not seem to preclude the proposed "question type x word type" interaction, which is driven by priming (according to the dual-route model). In a control analysis, we did not find evidence for an effect of word frequency on RTs in our data set[4].

As *Sparrow (2018)* in her reply to the replication strongly argues against presenting each target word more than once (referring to "active thought suppression"), we decided to randomly select words for the easy and hard blocks. From our pool of 48 words, eight computer terms and 16 general terms were presented in each block, so that each word was presented only once in the experiment. The words were randomly chosen for each participant. Of note, each word was presented twice in the original study, once in the easy block, and once in the hard block. Since we did not intend to select single words for post hoc pairwise tests, we did consider a strict "no word repetition" approach to be more in line with Sparrow's (2018) suggestions. Table 1 summarizes the known differences between the original study and the replication studies.

## Data preprocessing

We preregistered our analyses including confirmatory and exploratory statistical tests (https://aspredicted.org/z3xt4.pdf), and later decided to restrict the analysis to confirmatory tests. All data are available on OSF (https://osf.io/cjgea/). Data were preprocessed and analyzed using R version 3.3.2 (http://www.r-project.org), and RStudio version 1.0.136 (http://www.rstudio.com). The original "csv" data files were exported from oTree, imported into R, converted into the long format and merged using custom R scripts. Each student experimenter contributed one "csv" file containing the data from multiple participants. The resulting data file in the long format thus contains blocks of data from different student experimenters, because we did not sort the data according to recording time and date.

The age distribution of participants turned out to be heavily skewed (mean age: 31 years; range: 18–73). This was because the participant sample included students' families and friends, and we originally did not set a maximum age. Since the sample in the original paper (*Sparrow, Liu & Wegner, 2011*) consisted of undergraduate students (supplement, p.2), we decided to deviate from our preregistered protocol, and set the maximum age as the 75% quantile of the age distribution (44 years). In the remaining sample of 89 participants, the mean age was 24 years.

A second deviation from the preregistered protocol was due to the fact that we originally assumed that trials with incorrect responses (i.e., when participants pressed the key for the wrong color) should be excluded from the analysis. According to the replication report (https://osf.io/84fyw/), which includes personal communication with the original authors, this was not the case in the original study. We therefore included trials with correct and incorrect responses in the color-naming task. (Note that trials with incorrect responses in the memory task were not excluded, either.)

Although not specified in the original study, and neither in the replication study, we preregistered an additional exclusion of trials with RT outliers and anticipatory responses

Hesselmann (2020), *PeerJ*, DOI 10.7717/peerj.10325

**Table 1** **Differences between the original study by *Sparrow, Liu & Wegner (2011)* and the replication studies.**

| | *Sparrow, Liu & Wegner (2011)*, **Exp.1** | *Camerer et al. (2018)* | **This study** |
|---|---|---|---|
| Data collection | 2006 | 2017 | 2019 |
| Presentation software | DirectRT | oTree | oTree |
| Participant sample | Undergraduate students (USA) | Students and non-students (USA) | Students and non-students (Germany) |
| Sample size | $N = 46$[a] | $N = 104$ (+130)[c] | $N = 117$ |
| Trivia questions | Original set (16 easy, 16 hard) | Original set (16 easy, 16 hard) | Revised & translated set (16 easy, 16 hard) |
| Stroop words | Original set (incl. 16 internet words) | Original subset (incl. 8 internet words) | Revised, translated & validated set (incl. 16 internet words) |
| Number memory task | On each Stroop trial | 1× per block of trials (easy/hard) | On each Stroop trial |
| Number of presentations per Stroop word | 1× per block of trials (easy/hard)[b] | 2× per block of trials (easy/hard) | 1× per experiment |
| Total number of trials | 48 (24 easy, 24 hard)[b] | 96 (48 easy, 48 hard) | 48 (24 easy, 24 hard) |
| Participant debriefing | Unknown | Unknown | Yes |

**Notes.**

[a] Original sample size is based on the supplementary information.

[b] Total number of trials is an informed guess based on the original study and the response to the failed replication (*Sparrow, 2018*).

[c] The authors used a two-stage procedure for the replication. Stage 1 had $N = 104$, and stage 2 had $N = 234$. First and second data collections were pooled.

(<0.1s). We defined RT outliers using the interquartile range (IQR) method (*Tukey, 1977*), separately for each participant. These criteria resulted in the exclusion of $7 \pm 3\%$ of trials (mean percentage $\pm$ standard deviation). For our final analysis, we did not exclude RT outliers, but also report the results of the analysis including the exclusion of RT outliers.

## Data analysis

For each participant, we calculated the performance in the easy and hard questionnaires, the performance in the number memory task, and the mean RT in each of the four conditions. The condition averages from all participants were then exported into JASP 0.10.2 (https://jasp-stats.org/) for frequentist and Bayes Factor (BF) analysis, using default Cauchy priors (scale 0.707). To test the predicted "question type $\times$ word type" interaction, we calculated the following difference and tested it against zero using a two-sided paired test: $[RT(computer) - RT(general)]_{hard} - [RT(computer) - RT(general)]_{easy}$. Performance in the number memory task was calculated for all trials (i.e., including trials with RT outliers in the color-naming task).

## RESULTS

Participants found the easy questions to be answerable ($97 \pm 4\%$, mean accuracy $\pm$ standard deviation), but had difficulty finding the correct answers to the hard questions ($60 \pm 11\%$). As in the original study (98% versus 47%), this difference was significant ($t_{88} = 48.31$, $p < .001$). Mean accuracies in the color-naming task and number memory task were high ($98 \pm 3\%$ and $81 \pm 16\%$, respectively).

Figure 2A plots the RT data from the color-naming task. The dual-route model predicts that computer terms create more interference, and thus are associated with longer RTs than general terms, in particular in hard question blocks. The results show that in easy question blocks, mean RT was $830 \pm 366$ ms for general terms, and $901 \pm 620$ ms for computer terms (mean $\pm$ standard deviation). In hard question blocks, mean RT was $825 \pm 453$ ms for general terms, and $821 \pm 379$ ms for computer terms. Thus, the RT data do not show the predicted pattern. Accordingly, the sequential Bayes Factor (BF) analysis yielded a final $BF_{01}$ of 5.07 in favor of the null over the alternative hypothesis, after the data from $N = 89$ participants were taken into account (Fig. 2B). The observed data are thus 5 times more likely under the null model than under the alternative model (i.e., the "question type x word type" interaction). When the specific data pattern reported in the original study was considered as alternative model in an exploratory analysis, our data were 16 times more likely under the null model ($BF_{0+} = 16.43$). The preregistered two-sided one-sample $t$-test was not significant ($t_{88} = -1.04$, $p = .301$).

The pattern of results remained the same when incorrect responses in the color-naming task were excluded. When RT outliers were excluded from data analysis, overall mean RTs decreased, but the predicted interaction could still not be observed ($BF_{01} = 3.74$; $t_{88} = 1.31$, $p = .194$). In easy question blocks, mean RT was $737 \pm 296$ ms for general terms, and $723 \pm 295$ ms for computer terms (mean $\pm$ standard deviation). In hard question blocks, mean RT was $707 \pm 268$ ms for general terms, and $712 \pm 261$ ms for computer terms (data not plotted in a figure).
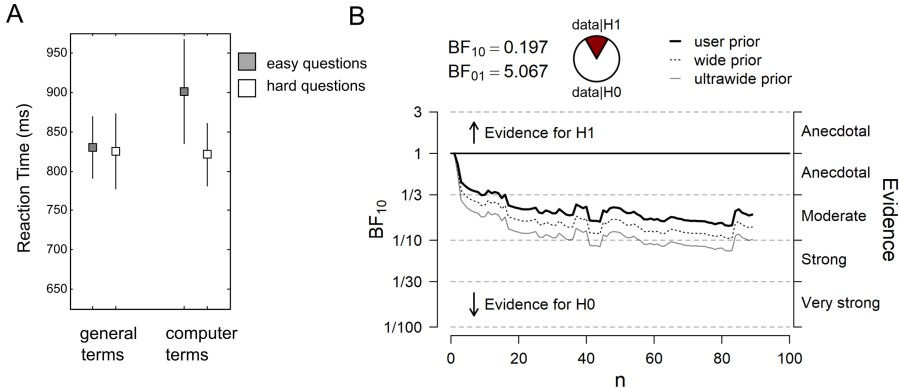

**Figure 2  Main results ($N = 89$).** (A) Reaction time (RT) data from the color-naming task. Grey squares represent the mean RT in the "easy question" blocks, white squares represent the mean RT in the "hard question" blocks. Errors bars represent the standard error of the mean. (B) Sequential Bayes Factor (BF) analysis of the "question type × word type" interaction. User prior refers to a Cauchy prior with scale 0.707, wide prior to a Cauchy prior with scale 1, and ultrawide prior to a Cauchy prior with scale $\sqrt{2}$.

In the debriefing after the main experiment, 11 out of 89 participants did not provide a response (e.g., closed the browser before answering). 73% (57/78) of all responders said that they would look up the answer on the Internet when faced with a hard general knowledge question in their daily life; 18% (14/78) said that they would ask someone who might know the answer, and 9% (7/78) responded that they would "leave it at that". When we restricted the RT data analysis to participants consulting the Internet ($N = 57$), the pattern of results was very similar to the one reported above ($\text{BF}_{01} = 5.01$). Finally, when all participants ($N = 117$) were included in the data analysis, the pattern of results turned out to be 9 times more likely under the null model than under the alternative model ($\text{BF}_{01} = 9.31$).

## Statistical issues in the original study

During our work on this project, we noticed three statistical issues in the original paper by *Sparrow, Liu & Wegner (2011)*. In the following paragraph, we address these observations in turn.

First, it remains unclear whether $N = 69$ or $N = 46$ participants were tested in Experiment 1 of the original paper (*Sparrow, Liu & Wegner, 2011*). In the supporting online material https://science.sciencemag.org/content/suppl/2011/07/13/science.1207745.DC1 the authors write that "Forty-six undergraduate students (28 female, 18 male) at Harvard University were tested in a within subjects experiment" (p.2). In their 2 × 2 within-subject design (easy/hard questions; computer/general words), the correct degrees of freedom (df) would then be 45 for paired $t$-tests, as well as for main effects, the interaction and simple main effects in the rm-ANOVA. Under the assumption that $N = 69$ participants were tested, the correct df would be 68. In the main text and supplement of the original paper, the reported df is 68 for paired $t$-tests, and 66 for the rm-ANOVA. According to the authors of the replication, the original authors initially confirmed that the sample size was 46 participants, and that dfs were misreported in the paper, but after the publication

[5]In the main text of *Sparrow, Liu & Wegner (2011)*, the simple effect is reported as follows: F(1,66) = 4.44, $p < .04$ (exact $p$-value: .03891. For $N = 46$, the result would be: F(1,45) = 4.44, $p = .040712$, and thus $p > .04$.

of the replication the original authors pointed out that the number of participants was 69 (https://osf.io/84fyw/). While the reported t-, F-, and $p$-values appear to be in better agreement with the $N = 69$ scenario[5], the reported dfs are incorrect in both scenarios.

Second, two computer-related words (Google/Yahoo), and two unrelated words (Target/Nike) were selected for analysis in the original paper. In the main text, RT data are reported only for this subset, together with the "question type x word type" interaction (F(1,66) = 5.02, $p < .03$). The reasoning behind this post hoc selection (i.e., selection of four words out of a pool of Stroop words) remains unclear. Without further information, it remains possible that the choice was made after seeing the data. Such post hoc choices and data-contingent analyses are misleading when they are not presented as exploratory analysis. The potential impact of this post hoc selection should not be underestimated. As pointed out by Sparrow in a reply to the failed replication, "each Stroop word was seen only once by participants" (*Sparrow, 2018*). Hence, the RTs for Google/Yahoo and Target/Nike reported in the original paper are based on single trials per participant. Given such noisy measurements with low precision (*Smith & Little, 2018*), the authors might have capitalized on chance by selecting a subset of words for their main analysis.

Third, two paired $t$-tests on word type are reported for the complete data set ("hard questions" condition, computer words versus general words: $t(68) = 3.26$, $p < .003$; "easy questions" condition, $t(68) = 2.98$, $p < .005$). The "question type ×word type" interaction is reported only for the 4-word-subset (F(1,66) = 5.02, $p < .03$). Based on visual inspection of the reported average RTs, it seems plausible that the "question type x word type" interaction should be smaller for the complete data set (Fig. 3A) than for the 4-word-subset (Fig. 3B). *Sparrow, Liu & Wegner (2011)* report the following result in the supplement: "Taking out the 4 terms (Google/Yahoo and Target/Nike) which yielded an interaction with easy/hard questions (F(1,66) = 5.52[sic], $p < .03$), the interaction between computer and general terms and easy/hard questions *remains* significant F(1,66) = 9.49, $p < .004$" (p.3, italics added). In fact, the interaction for the reduced data set (i.e., all data minus the 4-word-subset) turned out to be *larger* than the interaction reported for the subset (F-values 9.49 and 5.52, respectively). This result seems difficult to reconcile with the reported data, but access to the original data would be necessary to clarify this point.

## DISCUSSION

Although the majority of participants in our study reported that they would normally look up the answer to hard general knowledge questions on the Internet, we did not find evidence for the "Google Stroop effect", as originally published by *Sparrow, Liu & Wegner (2011)*. Thus, our data are more in line with the results from *Camerer et al. (2018)* who failed to replicate the original effect in two independent experiments. Importantly, the design of our study considered the suggestions for improvement provided by *Sparrow (2018)* in response to the failed replications. Based on her commentary, we carefully updated and validated the computer-related terms, strictly avoided word repetitions, and manipulated cognitive load as in the original study.

It seems worthwhile to take a closer look at the hypothetical cognitive model underlying the "Google Stroop effect". *Sparrow, Liu & Wegner (2011)* describe their Exp.1 as a

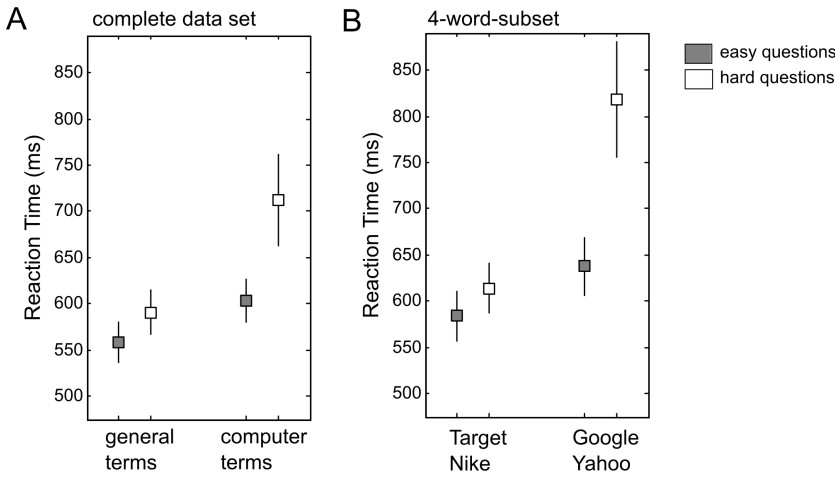

**Figure 3** **Reaction time (RT) data from Exp.1 in** *Sparrow, Liu & Wegner (2011)*. (A) RTs for the complete data set (16 general words, 8 computer-related words). (B) RTs for the 4-word-subset (2 general words: Target and Nike; 2 computer-related words: Google and Yahoo). Grey squares represent the mean RT in the "easy question" condition, white squares represent the mean RT in the "hard question" condition. Errors bars represent the standard error of the mean (for $N = 69$).

modified Stroop task, as follows: "People who have been disposed to think about a certain topic typically show slowed reaction times (RTs) for naming the color of the word when the word itself is of interest and is more accessible, because the word captures attention and interferes with the fastest possible color naming" (p.776). Classic and modified Stroop tasks, such as the emotional Stroop task, have been discussed in much detail elsewhere (*MacLeod, 1991*; *Algom, Chajut & Lev, 2004*). In Exp.1 by *Sparrow, Liu & Wegner (2011)*, there are two colors (red, blue), and two types of words: general terms (e.g., sport) and computer terms (e.g., Google). The authors propose a priming mechanism that specifically affects response times in trials with computer terms: "not knowing the answer to general knowledge questions primes the need to search for the answer, and subsequently computer interference is particularly acute" (p.776).

According to this model, computer terms become more accessible when the concept of knowledge is activated, or when information necessary for answering a trivia question is lacking, which results in more interference. We think that at least two points need to be made about this model. First, what remains unspecified is the duration of the proposed priming effect (i.e., for how long the computer terms are more accessible than general terms due following the activation of concepts). Follow-up studies could focus more on the temporal dynamics of the proposed priming effect. Perceptual and semantic priming effects are typically short-lasting, within the range of hundreds of milliseconds, while priming effects from social psychology (e.g., the "Florida effect") are substantially longer lasting, but have turned out to be not robust (*Harris et al., 2013*; *Doyen et al., 2014*). Second, the exact role of cognitive load (i.e., working memory load) for the "Google Stroop effect" remains somewhat unclear. In her response to the failed replications, *Sparrow (2018)* links the working memory task to the paradigm of active thought suppression: "when

people are asked explicitly not to think about a single target word, they must engage in active suppression" (p.1). *Sparrow (2018)* cites an earlier study (*Wegner & Erber, 1992*) in which participants performed a color-naming task similar to Exp.1 from *Sparrow, Liu & Wegner (2011)*. In this experiment, strongest Stroop-like interference was observed when participants were suppressing a specific target word under cognitive load, and when they were asked to name the color of this target word. In contrast to the "Google Stroop effect", however, participants were asked to actively suppress a single word, so that the importance of high cognitive load in one task might not tell us much about the role of high cognitive load in the other task. Alternatively, it could be argued the word's meaning (in contrast to its color) acts as distracting information in the color-naming task, and that high working memory load increases distractor processing (*Lavie, 2005*). Therefore, the manipulation of cognitive load might be crucial for the processing along the word reading pathway, and thus for the emergence of the "Google Stroop effect".

Our data, however, do not support the notion that the concept of the Internet (together with computer-related terms) becomes automatically activated when participants need to answer difficult general knowledge questions. We are aware that our study does not provide a definite answer, and the potential effects of "cognitive offloading" on human cognition (*Risko & Gilbert, 2016*) are definitely worth further attention and investigation. As mentioned above, the original study by *Sparrow, Liu & Wegner (2011)* avoided word repetitions, so that the RT data from each participant were based on single presentations of target words. Given such noisy RT measurements, in combination with the suboptimal analysis of variance on the sample mean (*Whelan, 2008*), post hoc selection of target words can easily lead to the wrong impression that an effect exists (e.g., we might find a "Wikipedia Stroop effect" or "Firefox Stroop effect" in our data set). Therefore, we recommend using the complete data set in future studies when testing for the crucial "question type x word type" interaction, preferably with linear mixed-effects models that can better account for stimulus-driven variability in RTs than repeated measures ANOVA (*Baayen, Davidson & Bates, 2008*). Fitting more complex linear models, e.g., models with random slopes, would also require more data than in the present experimental design in which a limited set of words is presented only once per participant (*Meteyard & Davies, 2020*).

## CONCLUSION

Our results revealed no evidence in favor of the notion that the concept of the Internet or internet access (via computers or smartphones) becomes automatically activated whenever participants are faced with hard trivia questions. Thus, the "Google Stroop effect" might be much smaller than previously thought, and less robust to variations in the experimental design. What else have we learned from this second replication of the "Google Stroop effect"? Our work on this project revealed that the original research design might not be a good starting point to further investigate this effect. To study the effects of the digital "all-knowing cloud" on our cognition, research designs with more precision and power are clearly necessary. Finally, as has been so adequately pointed out by Camerer and colleagues, this set of replications "illustrates the importance of open access to all of

the materials of published studies for conducting direct replications and accumulating scientific knowledge'' (*Camerer et al., 2018*).

## ACKNOWLEDGEMENTS

This work is based on an undergraduate student research project (''Experimentell-Empirisches Praktikum—ExPra'') at the Psychologische Hochschule Berlin (PHB) in summer 2019. The following students took part in this project and collected the data: Franzceska Cubela, Luisa Engel, Isabella Ermel, Eran Fortus, Lena Jaeschke, Sarah Koch, Morgane Kroeger, Nikita Kühn, Fabio Kunze, Maximilian Kurtz, Clara Kursawe, Can-Pascal Scheftlein, Alexandra Seidel, and Nora Teichmann. We thank Leona Hammelrath for her invaluable support in helping us set up the oTree experiment and the R scripts, and Marcus Rothkirch for his helpful comments on an earlier version of the manuscript.

### Funding

The author received no funding for this work.

### Competing Interests

The author declares there are no competing interests.

### Author Contributions

- Guido Hesselmann conceived and designed the experiments, performed the experiments, analyzed the data, prepared figures and/or tables, authored or reviewed drafts of the paper, and approved the final draft.

### Human Ethics

The following information was supplied relating to ethical approvals (i.e., approving body and any reference numbers):

The ethics committee of the Psychologische Hochschule Berlin (PHB) approved this study (PHB10032019).

### Data Availability

All data and materials are available at OSF: Hesselmann, Guido. 2020. ''Replication of the 'Google Stroop Effect.''' OSF. August 12. https://osf.io/cjgea.

### Supplemental Information

Supplemental information for this article can be found online at http://dx.doi.org/10.7717/peerj.10325#supplemental-information.

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
