# Peer review of "No conclusive evidence that difficult general knowledge questions cause a “Google Stroop effect”. A replication study"

_PeerJ, doi:10.7717/peerj.10325_

## Round 0.1 · original submission · Minor Revisions

Your manuscript has now been seen by two reviewers. You will see from the comments below that some constructive points are worth considering. We therefore invite you to revise and resubmit your manuscript, taking into account these points. Please highlight all changes in the manuscript text file.

·

Basic reporting

No comment.

Experimental design

No comment.

Validity of the findings

No comment.

Additional comments

GENERAL EVALUATION
This is a very straight-to-the-point and transparent manuscript describing the much-needed replication of a high impact cognitive psychology experiment. In recent years, many high-impact findings that describe implicit/unconscious effects of observers' cognitive states on subsequent perception and behavior have appeared to be less robust that initially believed, following failed replications. In this particular case, a previous replication that failed to replicate the effect-of-interest of the original study, differed from the original study in a number of potentially crucial ways (as pointed out by one of the original study's authors) due to missing information at the time of the previous replication. The current study has been set up with care to create a replication of the original experiment that satisfies all the recommendations and requirements laid out by the original authors, and constitutes a near-identical replication of the original study. This is important work, not only because it provides substantial evidence against a high-impact effect-of-interest, but also for its exemplary role in promoting transparent and robust science. I have only a number of minor suggestions, which are listed below.

STRENGTHS
o All stimulus materials were pre-tested and validated (i.e., internet-related words were truly internet related, and significantly more internet related than general words, which were not; difficult tasks were truly more difficult than easy tasks, and difficulty was comparable to that of the original study; the current sample of participants confirmed having the urge to look up answers to difficult questions online; generally speaking, all sanity checks were conducted and were satisfactory).
o All analysis and design choices were clearly and transparently motivated, in relation to the original study, the replication, the original author’s response to this replication, and the preregistration of the current experiment.
o The manuscript ends with sound recommendations for analyzing the current type of data in future projects (using linear mixed effects models, better suited to the current data type), and with a number of caveats in the analysis/design of the original study that helps the readership to weigh the evidence provided of the original study as compared to that of the current (and previous) replication.

SUGGESTIONS FOR CLARIFICATION OR IMPROVEMENT
o Line 213 mentions that the author decided to skip the preregistered exploratory tests. Why?
o Lines 227-233: in line with the original study, response times to incorrect RTs were analyzed as well. To me (and to the author as well, apparently), this seems very strange. Could the author perhaps confirm in the text body (or a footnote) whether or not the pattern of results is similar when only correct responses are analyzed? Somewhat related to this, was the accuracy on the color-naming task comparable to that of the original study?
o Lines 265-266 states that the current data were 5 times more likely under the null model than under THE alternative model. But the correct way to phrase this, given the statistical test that was conducted, is that the current data were 5 times more likely under the null model than under ANY alternative model (i.e., it’s a two-sided test). Of course, the author needed to conduct this two-sided test because it was pre-registered, but it would be valuable to also add the Bayes Factor for the null model relative to THE (i.e., expected) alternative model (i.e., BF0+), which shows how much more likely the null is compared to the specific pattern of findings observed in the original study. (Please do mention explicitly that this is a non-registered, exploratory analysis.)
o Lines 273-277 describes that most participants (57 in total) would use the internet to look up the answer to difficult questions. As such, it could be worthwhile to explore whether the observed pattern of data (i.e., the null effect) also holds for this specific subgroup of participants.
o Lines 385 mentions “the effect” in “to what degree the effect depends on cognitive load”, but based on both the current replication and the previous one, there is no effect.
o In the Results and General Discussion sections, the author – rightfully – mentions the caveat of post-hoc selection of a subset of the data for making inferences from the data (as was done in the original study, by presenting the results of 4 words out of the full stimulus set). To drive the point home, it could be valuable to show examples of a selection of 4 words for which such an effect would have been statistically significant in the present dataset (if any), and/or quantify how many random selections of 4 words would have yielded any pattern of significant effects.

Reviewer 2 ·

Basic reporting

See main review.

Experimental design

See main review.

Validity of the findings

See main review.

Additional comments

The author reports the results of a direct replication of a priming study by Sparrow et al. (2011). A previous attempt to replicate the Sparrow et al. finding failed (Camerer et al., 2018), but the lead author of the original study (Sparrow) argued that the previous replication attempt was not a *direct* replication. The present replication attempt was much closer to the original study, but again the finding was negative.

I support the publication of direct replications, regardless of whether the replication is successful or not. So, for me, the question is whether this study is a direct replication. I think it is close enough, though I would argue that the author needs to carefully specify the differences between the present study and the original study (see #1). I have other concerns as well, though I believe these concerns can be addressed through revision.

1. I struggled to understand the differences between the method used in the present study and the method used in the original study. I suggest that the author list every known difference in clear, concrete language, perhaps in a numbered list.

2. On lines 222 through 226, the author writes that he removed older subjects from the sample because the sample in the original study included only undergraduate students, and this exclusion criterion was not listed in the preregistration. The author deserves kudos for making this disclosure. Still, I think readers need to see both version of the results: that is, a) with the older subjects, and b) without the older subjects. In fact, maybe the author could add a figure with three panels: one showing the results of the original study, one showing the results of the present study with all subjects, and showing the results of the present study without the older subjects.

3. I don’t think the present study needs a theoretical story, especially since the observed finding is a null result, so I would recommend omitting Figure 4 and much of the discussion. I think it makes more sense to frame the study as a replication attempt.

---

## Round 0.2 · accepted · Accept

Thank you for the revised manuscript and response letter. I am pleased to inform you that your manuscript has been accepted for publication.

·

Basic reporting

no comment

Experimental design

no comment

Validity of the findings

no comment

Additional comments

The author has satisfactorily addressed all issues raised during the previous review round. I have no further concerns.

Kind regards,

Surya Gayet

Reviewer 2 ·

Basic reporting

Writing is very clear. Figures are informative, and Figure 2 is one of the best figures I've seen.

Experimental design

Design is straightforward. Replication is very close to the original, and the differences between the present study and the original are minor and also clearly listed in Table 1.

Validity of the findings

The findings are clear. The Bayes Factor values make it clear that the data provide no good evidence for the claim of the original study by Sparrow et al.

Additional comments

I thought the revised manuscript was clear, and the author was quite responsive to my comments (he added Table 1 and included the results of an additional analysis). The data collection and analyses appear to be first rate, and I recommend publication.